# Effectiveness of a universal digital–human parenting intervention in promoting early childhood development and protection: A pragmatic cluster randomized controlled trial

Zuyi Fang[1,2‡]*, Qing Han[1,2‡], Ruochen Ruan[3], Xinyu Shi[4], Cheng Zhang[5,6], Dongqin Ruan[7], Xiangming Fang[8,9], Inge Vallance[2], Jamie M. Lachman[2,10,11]

**1** Institute of Population Research, Peking University, Beijing, China, **2** Department of Social Policy and Intervention, University of Oxford, Oxford, United Kingdom, **3** Faculty of Education, Beijing Normal University, Beijing, China, **4** School of Government, Beijing Normal University, Beijing, China, **5** School of Sociology and Social Policy, University of Leeds, Leeds, United Kingdom, **6** China Logistics Group, Beijing, China, **7** Chengbei Preschool, Xinyu, China, **8** College of Economics and Management, China Agricultural University, Beijing, China, **9** School of Public Health, Georgia State University, Atlanta, Georgia, United States of America, **10** Centre for Social Science Research, University of Cape Town, Cape Town, South Africa, **11** Parenting for Lifelong Health, Oxford, United Kingdom

‡ Co-first authors.
* fangzuyi@pku.edu.cn

**Editor:** Hongxing Luo, Maastricht University Cardiovascular Research Institute Maastricht: Universiteit Maastricht Cardiovascular Research Institute Maastricht, NETHERLANDS, KINGDOM OF THE

## Abstract

Delayed early childhood development and violence against children are major global challenges, particularly in low-resource settings. Universal digital–human parenting interventions may offer a scalable solution by overcoming barriers associated with traditional in-person programs. This study reports the first pragmatic randomized controlled trial evaluating a blended chatbot-based parenting intervention delivered within the Chinese preschool system. The trial was conducted in a lower-middle-income city in central China. Twenty-one preschool classes were cluster-randomized to a treatment group (n = 10) or waitlist control (n = 11). Primary caregivers of enrolled children participated following informed consent. The intervention comprised a 2.5-month chatbot-led digital parenting program supported by weekly or twice-weekly online group sessions facilitated by headteachers and social workers. Data were collected at baseline, post-intervention, and at 6- and 12-month follow-ups. Primary outcomes were caregiver-provided early learning and stimulation, and caregiver-perpetrated violence. Analyses followed the intention-to-treat principle using multi-level regression models. Equity effects related to caregiver and child disability were explored through moderation and subgroup analyses. Sustainability of impacts was assessed, and complier average causal effects examined the role of intervention completion. Between March 2024 and June 2025, 541 caregivers of children aged 3–6 years were enrolled (treatment: n = 272; control: n = 269), of whom 25.2% were male. Overall, 60.3% completed all chatbot modules. At post-intervention, the program significantly improved early learning and stimulation (β = 1.79, 95% CI [0.24,

**Data availability statement:** All relevant data are within the paper and its Supporting information files.

**Funding:** This study is part of the Global Parenting Initiative, which is funded by the LEGO Foundation and Oak Foundation. This research was supported by the Peking University Start-Up Research Grant (7101303781 to ZF) and the China Postdoctoral Science Foundation (2022M720467 to ZF). The funders had no role in study design, data collection and analysis, decision to publish, or preparation of the manuscript.

**Competing interests:** I have read the journal's policy and the authors of this manuscript have the following competing interests: JML is the CEO of Parenting for Lifelong Health (PLH), a charitable organization based in the United Kingdom that developed the adapted program. ZF and QH have worked as a consultant for PLH in the past. JML has (and is participating) in several research studies involving the program, as an investigator, and the University of Oxford, and University of Cape Town receive research funding for these.

3.34]) and reduced caregiver-perpetrated violence (IRR = 0.87, 95% CI [0.80, 0.96]). The intervention showed potential to advance equity for families affected by disability, with some effects sustained at follow-up. Complier analyses indicated reduced endorsement of corporal punishment and lower parental anxiety among participants completing at least 30 modules. Universal digital–human parenting interventions embedded in preschool systems can enhance early childhood development and reduce violence, highlighting the importance of human support and cultural adaptation to optimize engagement and outcomes.

## Author summary

Many young children around the world do not receive enough learning opportunities at home and are exposed to violent discipline, which can negatively affect their health and development. Parenting programs can help, but in many settings they are difficult to deliver because they require time, travel, and specialized staff. We wanted to examine whether digital tools, combined with limited human support, could provide a practical and inclusive alternative.

In this study, we tested a parenting program delivered mainly through a smartphone chatbot, supported by online group discussions led by preschool teachers and social workers. The program was offered to all families with preschool-aged children in a public preschool in China, allowing us to assess its effectiveness under real-world conditions.

We found that families who received the program spent more time supporting their children's learning and used less physical punishment. Parents also reported fewer emotional problems in their children and lower levels of anxiety themselves. These benefits were observed across diverse families, including those affected by disability, and some improvements lasted beyond the end of the program.

Our findings show that combining digital parenting support with human guidance can be a scalable way to promote healthy child development and reduce harm. This approach may be especially valuable in communities where access to traditional services is limited.

## Introduction

Delayed early childhood development (ECD) remains a global challenge. Approximately 250 million children under five worldwide are at risk of not reaching their developmental potential, with children in low- and middle-income countries (LMICs) and those with disabilities disproportionally affected [1]. In China, 17% of children in this age group—around 14 million—face such risks due to stunting (defined as height-for-age more than two standard deviations below the WHO Child Growth Standards median) or poverty [2]. Early childhood is a period when the brain is highly

sensitive to stimulation and care. It forms the foundation for lifelong development, psychological well-being, and economic productivity [3,4].

ECD is shaped by the dynamic interaction between the child and their environment, with nurturing care as a fundamental determinant of healthy psychosocial adjustment. This includes health and nutrition support, early learning opportunities, responsive caregiving, and safety and security [5]. Primary caregivers are the architects of this environment; without nurturing care, children miss critical developmental opportunities and face increased risks of harm, including violence, and subsequent psychopathology.

Violence against children (VAC) is not only a human rights violation but also a potent risk factor for psychiatric morbidity. Globally, around one billion children experienced VAC in the past year, with children with disabilities more than twice as likely to be affected [6,7]. In the Chinese Mainland, the lifetime prevalence of caregiver-perpetrated VAC is estimated at around 30% [8]. Exposure to VAC, particularly in early childhood, exacerbates developmental delays, precipitates the onset of internalizing and externalizing symptoms, and undermines educational and economic prospects [9].

ECD parenting programs aim to strengthen caregivers' ability to provide nurturing care and protect children from violence [5]. In-person programs have demonstrated positive effects on child development across multiple domains, and have altered factors associated with caregiver-perpetrated violence for families of children with and without disabilities [10,11]. Despite this evidence, scalability remains limited due to logistical (e.g., time, transport, childcare), structural (e.g., limited and costly human resources), and cultural (e.g., stigma around receiving mental health or parenting support) barriers [12].

Digital parenting interventions have emerged as a scalable alternative to in-person delivery [13]. Increasing global internet access and digital literacy have improved their reach and potential for universal public health impact. Meta-analyses from high-income countries show that digital parenting programs targeting high-risk populations can achieve multiple positive child and parental outcomes [14–21]. Although low-risk families also face challenges in supporting child development and protection—for example, a survey of nearly 200,000 families across nine Chinese provinces found that 84.5% of caregivers expressed a need for evidence-informed parenting support to address daily challenges [22]—few studies have tested universal intervention approaches. A recent meta-analysis identified 22 RCTs and quasi-experiments of self-guided, universal digital parenting programs, showing post-intervention improvements in parental mental health, self-efficacy, and social support [23]. Only three studies were conducted in LMICs, all with small sample sizes, and none examined disability inclusion.

Self-guided digital interventions often face low engagement in real-world settings, with completion rates ranging from 0.5% to 28.6%, limiting their therapeutic effectiveness and public health impact [24]. Human interaction is increasingly recognized as essential for establishing a therapeutic alliance and improving engagement in digital behavior-change interventions [25]. However, there is a lack of rigorous evaluations of universal parenting programs that combine digital and human components and assess outcomes beyond immediate post-intervention, particularly in LMICs.

This study aims to evaluate the effectiveness of a universal digital-human parenting intervention, embedded in the preschool setting, for Chinese caregivers of preschool-aged children. Specific objectives include enhancing early learning and stimulation, reducing caregiver-perpetrated violence, and mitigating child behavioral problems. It also examines the sustainability of effects at 6- and 12-month follow-up and explores equity impacts among caregivers and children with disabilities.

To our knowledge, this is the first study of its kind. Findings aim to inform the design and scale-up of effective and inclusive digital-human parenting interventions to address the universal reality of poor ECD and VAC, ultimately promoting population-level child mental health, especially in low-resource settings.

## Materials and methods

### Study design and participants

A pragmatic, two-arm, single-blind, cluster RCT with a waitlist control group was conducted. A cross-over design was not adopted due to the expected persistence of intervention effects, which would violate the assumption of no carryover between periods. A waitlist control was used to ensure all participants had access to the intervention if beneficial, and to

enhance recruitment and retention by assuring control families of eventual participation. The trial took place in Chengbei Preschool, one of the largest public preschools in Xinyu, a lower-middle-income city in Jiangxi Province, central China. The intervention was delivered as part of routine parenting support services.

Eligible participants were primary caregivers aged 18 years or older of children aged 3–6 years enrolled in the preschool, with access to a smartphone, who provided informed consent electronically after receiving an online information sheet and explanation of study procedures. As a universal intervention, no additional individual-level exclusion criteria were applied. All 21 classes within the preschool were eligible and included; no cluster-level exclusion criteria were applied. Headteachers invited all eligible caregivers within their respective classes through routine school communication channels.

No demographic or baseline data were collected from caregivers who declined participation, in order to respect privacy and ethical considerations. The trial protocol was prospectively registered on the Chinese Clinical Trial Registry (ChiCTR2400081911, https://www.chictr.org.cn/showproj.html?proj=221616)

### Randomization and masking

Chengbei Preschool comprises three grade levels, each with seven classes (21 classes in total). The class was the unit of cluster randomization. The preschool institution agreed to participate prior to trial initiation, and all 21 eligible classes were recruited before randomization. Institutional approval for cluster participation was provided by the preschool administration. Randomization was conducted after baseline caregiver consent was completed. Eligible classes were randomly allocated in a 1:1 ratio to the intervention group or the waitlist control group, using a computer-generated random sequence. To ensure allocation concealment, randomization was performed by an independent researcher not involved in the study design or implementation and with minimal subsequent involvement.

Outcome assessors and data analysts were blinded to group allocation, however, given the nature of the parenting intervention, blinding was not feasible for caregiver participants, program facilitators, preschool administrators, trial coordinators, or the principal investigator. All data collectors received standardized training in the study protocol and ethical procedures and were explicitly instructed to avoid seeking or inferring group assignment throughout the trial. Likewise, caregiver participants, facilitators, and trial coordinators were instructed not to disclose group allocation to assessors during data collection. There was no cluster-level crossover or noncompliance during the trial. All clusters randomized to the intervention arm received the intervention as allocated. Variation in adherence occurred only at the participant level.

### Intervention procedures

Headteachers recruited eligible caregivers from each class, with one caregiver per family. Trained data collectors obtained informed consent and conducted baseline assessments in March 2024 (T0) via videoconferencing, by administering encrypted online questionnaires and recording caregiver responses. Randomization was performed after baseline data collection.

The intervention, Keyushiguang, was adapted from Parenting for Lifelong Health's ParentText, a rule-based chatbot developed in collaboration with UNICEF, the University of Oxford, IDEMS International, and other academic and institutional partners, and made available as an open-access resource under a Creative Commons Attribution Share-Alike 4.0 license. The adapted version was delivered via WeChat and comprised 37–39 (depending on child age) brief interactive modules covering eight positive parenting topics over 2.5 months. It included daily chatbot-delivered content and weekly or twice-weekly WeChat message-based group discussions facilitated by social workers and headteachers. Chatbot content was personalized based on caregiver and child characteristics. Full details of cultural adaptation, module content, delivery format, and support features are provided in Table 1 in S1 Text.

Given that caregiver participation was central to intervention delivery, human elements were intentionally designed to enhance engagement. These human supports operated across institutional, relational, and motivational levels,

while serving four core engagement functions: accountability, social reinforcement, guided reflection, and behavioral reinforcement.

At the institutional level, the program was introduced and endorsed by headteachers, who maintained established and trusted relationships with families. Positioning the intervention within routine preschool parenting support services helped legitimize participation, reduce stigma, and normalize engagement within existing school-family structures.

At the relational level, weekly or twice-weekly message-based group discussions were facilitated by headteachers and social workers in existing WeChat class groups. These sessions aligned with ongoing chatbot topics and fostered peer exchange, collective participation, and social accountability. Facilitators used structured prompts and suggested scripts from a standardized manual to guide reflection, clarify misunderstandings, and contextualize digital content within caregivers' everyday parenting experiences.

At the motivational level, a child-centered token incentive mechanism was implemented. Children received small developmentally appropriate rewards linked to caregiver engagement with chatbot modules, indirectly reinforcing sustained participation through family-level motivation. In addition, facilitators provided regular reminders and encouragement through established communication channels to prompt timely module completion and maintain engagement momentum. Together, these human elements were integrated to enhance uptake, support adherence, and minimize attrition under real-world implementation conditions.

Regarding training and supervision, headteachers and social workers received a 6-hour structured entry training prior to intervention delivery. Training covered the theoretical foundations of the program, facilitation principles, use of the digital platform, and ethical considerations. Facilitators were provided with a standardized implementation manual, including structured guidance and suggested scripts for group interactions to support consistency across classes. During the intervention period, weekly supervision sessions were conducted to monitor progress and address implementation challenges. Ongoing support was available as needed to ensure procedural consistency and fidelity. Although minor variations in facilitation style were expected in this pragmatic setting, core intervention content was standardized through chatbot-delivered modules and structured discussion themes.

The intervention group began the program in April 2024, while the control group received no formal support during this period. Post-intervention assessments (T1) were conducted in June–July 2024 for both groups. The waitlist control group then began the program. A 6-month follow-up (T2) was conducted in December 2024 among the original intervention group, as the waitlist control group began receiving the intervention after T1. A 12-month follow-up (T3) with the same group was completed in June-July 2025.

Ethics approval was granted by Beijing Normal University (SSDPP-HSC-2024003) and the University of Oxford (SPI DREC 25 006). Informed consent was also obtained from all caregivers, headteachers and social workers, who were trained as program facilitators. All data were collected via encrypted online questionnaires and transmitted through secure connections. Survey responses were stored on password-protected servers with access restricted to authorized research personnel. Identifiable information was stored separately from survey data, and only de-identified datasets were used for analysis.

## Outcomes

The primary outcomes were child early learning and stimulation, measured using six items from the Multiple Indicator Cluster Surveys (MICS), and caregiver-perpetrated violence, assessed using nine items from the International Society for the Prevention of Child Abuse and Neglect Child Abuse Screening Tool for Trials (ICAST-T).

The secondary outcomes included child behavior problems, measured by parent reports on the Strengths and Difficulties Questionnaire (SDQ); parental mental health, assessed using the self-report Depression and Anxiety subscales of the Depression, Anxiety, and Stress Scale-21 (DASS); parenting stress, measured by the self-report Parental Stress Scale (PSS); positive parenting, assessed using the Alabama Parenting Questionnaire (APQ) Positive Parenting and

Involvement subscales; family function, measured with the Family APGAR scale; and attitudes toward corporal punishment, assessed with one item from the MICS survey. All instruments used in this study have been translated and psychometrically evaluated in Chinese populations, with evidence supporting their reliability and construct validity in family and child development research contexts. More details and references are presented in Table 2 in S1 Text.

All outcome measures were collected via caregiver self-report using encrypted online questionnaires administered by trained data collectors via videoconferencing. Participants were assured of confidentiality, and responses were not shared with preschool staff to reduce social desirability bias.

## Statistical analysis

**Intervention effect estimation.** All analyses followed the intention-to-treat principle, with participants analyzed according to their original group assignments, regardless of intervention completion or dosage received. Caregivers were free to decline participation before starting the survey. During survey administration, data collectors asked caregivers to answer all items and to flag any questions they found unclear or awkward; when this occurred, caregivers discussed their interpretation with the data collector (or referred the issue to the research team if necessary) and then provided the most appropriate response based on their understanding. As a result, item-level missing data were negligible by design. Any missing outcome data resulted from participant attrition at follow-up assessments and were handled using multilevel mixed-effects models estimated via maximum likelihood under standard missing-at-random assumptions. No multiple imputation was performed. Intervention effects were evaluated at post-intervention (T1).

The study design aimed to maximize use of available participant resources within the existing intervention delivery platform; therefore, the sample size was relatively fixed. Power calculations were conducted prior to implementation to estimate the minimum detectable effect size (MDES) for the primary outcomes under the fixed design constraints of the trial. Calculations assumed 21 clusters with 20–35 participants per cluster and intra-cluster correlation coefficients (ICCs) ranging from 0.01 to 0.05, reflecting values commonly reported in preschool-based and parenting intervention cluster trials. A two-sided significance level of $\alpha = 0.05$ was applied.

Under these assumptions, the MDES for Early Learning and Stimulation (linear multilevel model) ranged from $\beta = 0.12$ to 0.19 depending on cluster size and ICC values. For Caregiver-Perpetrated Violence (Poisson/negative binomial multilevel models), the MDES ranged from IRR = 0.83 to 0.89. Full details of the power calculations are provided in Table 3a and Table 3b in S1 Text.

For the frequency-based outcome of caregiver-perpetrated violence, Poisson or negative binomial regression models were applied, depending on the presence of overdispersion. Continuous outcomes were analyzed using multilevel linear regression models. All models specified participant ID nested within cluster (class) as random effects to account for between-class variability, including potential differences in headteachers and facilitation dynamics, and included fixed effects for group (intervention vs control), time (T0, T1, T2, T3; primary inference for between-group effectiveness was restricted to T1), and their interaction (group × time). Sensitivity analyses were conducted by adjusting for relevant covariates, including child age and gender, and caregiver age and gender.

To examine whether intervention effects differed by caregiver or child disability status, subgroup and moderation analyses were conducted by adding three-way interaction terms (group × time × caregiver/child disability) to the original models.

**Follow-up analysis in the intervention group.** Given the waitlist control design, the control group began receiving the intervention immediately after the post-intervention assessment (T1). As such, they could not be followed as a comparison group at the 6-month (T2) and 12-month (T3) follow-ups. Consequently, sustained intervention effects could not be evaluated through between-group comparisons over time.

To examine within-group changes in the intervention group, we conducted follow-up analyses at T1, T2, and T3. Generalized linear mixed-effects models were applied to intervention group data, using baseline (T0) as the reference.

Participant ID nested within cluster was included as a random effect, and time was treated as a fixed effect to assess the maintenance or change of outcomes over time within the intervention group.

**Causal effect among compliers and predictors of engagement.** To account for non-compliance in estimating intervention outcomes, we used a two-stage regression approach to estimate the complier average causal effect (CACE), with actual compliance defined as completing 30 or more intervention modules. As no universal standard exists, compliance was defined as completing approximately 80% of modules, consistent with thresholds (75–95%) used in prior studies. In addition, we examined potential baseline predictors of compliance, including caregivers' baseline scores on primary and secondary outcomes, caregiver and child gender and age, and cluster.

In addition to the pre-specified ITT and CACE analyses, a per-protocol sensitivity analysis was conducted. Participants in the intervention group who completed fewer than 30 chatbot modules were excluded, consistent with the compliance threshold defined in the Statistical Analysis Plan. Multilevel models identical to those used in the primary ITT analyses were applied to the restricted dataset.

This study is reported in accordance with the CONSORT 2025 statement and the CONSORT extensions for cluster randomized trials and pragmatic trials [26–28]. Reporting has been aligned with the recommended nomenclature and organization of study components.

## Results

A total of 665 eligible caregivers were invited to participate in the study, of whom 541 provided informed consent and completed the baseline assessment (T0), yielding an enrolment rate of 81.35%. Among the 21 classes, 10 clusters (n = 272) were randomly assigned to the intervention group and 11 clusters (n = 269) to the waitlist control.

At the subsequent three timepoints (T1, T2, and T3), 39, 59, and 114 caregivers did not complete the assessments due to time constraints, resulting in attrition rates of 7.2%, 21.7%, and 41.9%, respectively. All 21 randomized clusters (10 intervention, 11 control) were included in the primary analysis. The participant flow is presented in Fig 1.

Caregivers had a mean age of 36.6 years (SD 5.4). 74.8% (n = 404) were female and 25.2% (n = 136) were male. A small proportion (n = 4, 0.8%) were grandparents or other relatives. Most identified as Han ethnicity (n = 538, 99.4%). Around one-third (n = 177, 32.7%) had a rural household registration. The majority were married (n = 526, 98.5%). Most were employed full-time (n = 419, 80.4%), and 45 (8.6%) were self-employed. 37.5% (n = 203) had not completed an undergraduate education. Frequent use of digital devices was reported by 64.7% (n = 350), while 34.0% (n = 184) used them occasionally and 1.3% (n = 7) rarely. A total of 86 caregivers (15.9%) reported disabilities related to mobility, hearing, vision, adaptive living, communication, cognition, behavior, or other chronic conditions.

Children had a mean age of 4.9 years (SD = 0.9), and 43.5% (n = 234) were female. There were 1.9 children (SD = 0.6) and 4.5 members (SD = 1.1) living in the households on average. Mothers were the primary caregivers in 76.3% (n = 413) of families, followed by grandparents (n = 253, 46.8%) and fathers (n = 215, 39.7%). Disabilities were reported by caregivers in 23.7% (n = 128) of the children. As all clusters were within a single preschool, cluster-level characteristics were homogeneous and therefore not separately tabulated. Baseline demographic characteristics and outcome measures are presented in Table 4 in S1 Text.

### Program delivery

Among the 272 caregivers in the intervention group, 60.3% (n = 164) completed all chatbot modules. The average number of modules completed per person was 27.47 (out of 37 or 39, depending on child age). The module completion rate for caregivers who initiated interaction with the modules was 99.8%. The overall completion rate for each of the eight parenting topics ranged from 63.6% to 77.9%. When considering only caregivers who started at least one module with the topic, topic completion rates ranged from 81.5% to 96.7% (Table 5 in S1 Text). Fig 1 in S1 Text presents the distribution of total modules completed.

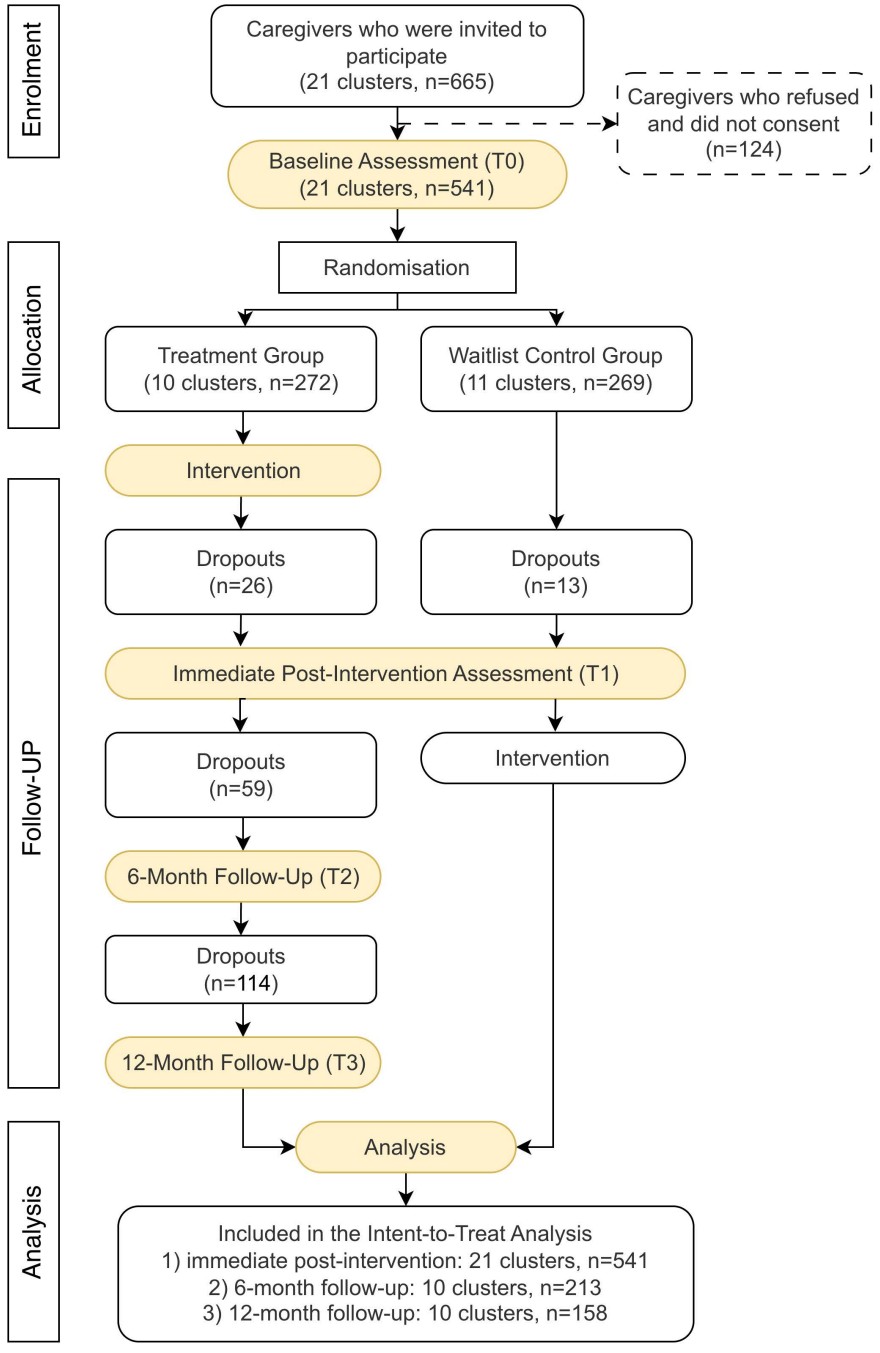

**Fig 1. Participant flow chart.** *Note*. At each follow-up, all participants originally allocated to the intervention group were invited to complete the assessment, regardless of prior participation. Dropout numbers were specific to each phase and not cumulative from earlier time points.

## Intervention effect estimation

Means and standard deviations for primary and secondary outcomes at T0 (baseline) and T1 (post-intervention) for both groups, along with summaries of missing data, are presented in Table 6 in S1 Text. Fig 2 visualizes both group-level and

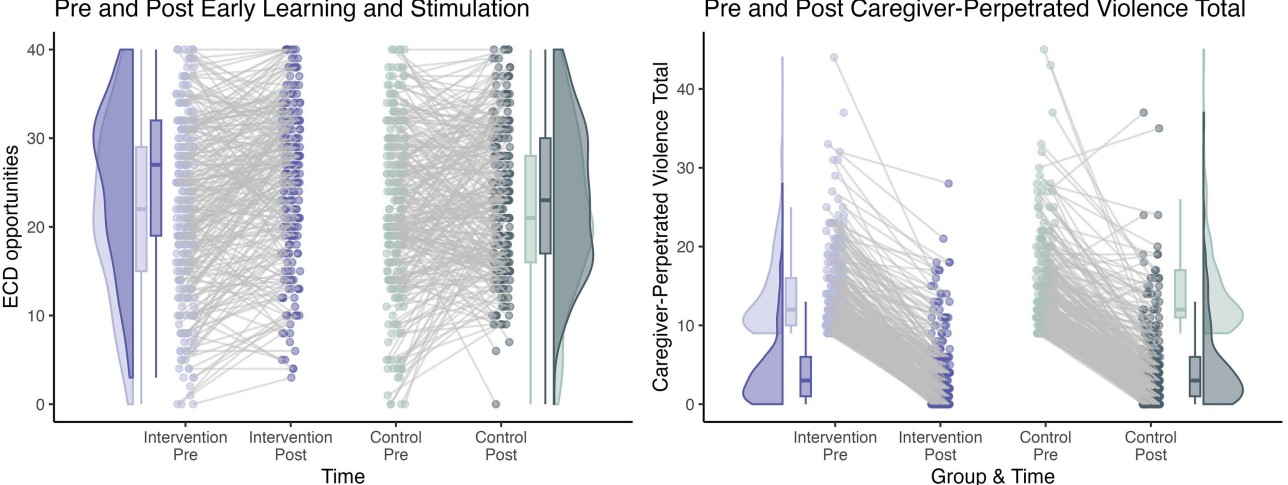

**Fig 2. Pre- and post-intervention changes in early learning and stimulation and caregiver-perpetrated violence.** *Note.* Each dot represents an individual participant's score at a given time point. Gray lines connect repeated measurements from the same participant to indicate within-person change. Violin plots and boxplots reflect the distribution (density, median, and interquartile range) of scores within each condition.

individual-level changes in primary outcomes from pre- to post-intervention, capturing overall trends and within-person variability. Multilevel regression models were employed based on distributional diagnostics (Table 7 in S1 Text).

At post-intervention, the treatment group demonstrated significantly greater early learning and stimulation compared to the control group (β = 1.79, 95%CI [0.24, 3.34], p = 0.023). Significantly lower levels of total caregiver-perpetrated violence (IRR = 0.87, 95%CI [0.80, 0.96], p = 0.005) and caregiver-perpetrated physical violence (IRR = 0.71, 95%CI [0.61, 0.84], p < 0.001) were also observed in the intervention group. No significant group difference was found for caregiver-perpetrated emotional violence (p = 0.521; Table 1).

Caregiver-Perpetrated Violence: Total was analyzed using a negative binomial multilevel regression model due to overdispersion. Caregiver-Perpetrated Violence: Physical and Emotional were analyzed using Poisson multilevel regression models. All other outcomes were analyzed using linear mixed-effects models.

Per-protocol analyses yielded results consistent with the ITT findings. Among participants who completed at least 30 modules, the intervention effect on Early Learning and Stimulation was β = 2.46 (95% CI [0.74, 4.17], p = 0.005), and the incidence rate ratio for Caregiver-Perpetrated Violence (Total) was 0.88 (95% CI [0.79, 0.99], p = 0.026) (see Table 8 in S1 Text).

For secondary outcomes, the intervention group showed significant reductions in support for corporal punishment (β = -0.29, 95%CI [-0.53, -0.06], p = 0.013), child internalizing behavior (β = -0.58, 95%CI [-0.97, -0.20], p = 0.003), and emotional problems (β = -0.42, 95%CI [-0.67, -0.17], p < 0.001), compared to the control group. No significant effects were observed for SDQ subscales for externalizing behavior, conduct problems, hyperactivity, or prosocial behavior. Significant improvements were noted in total parenting (β = 2.01, 95%CI [0.65, 3.37], p = 0.004), including positive parenting (β = 0.99, 95%CI [0.33, 1.66], p = 0.004) and parental involvement (β = 1.02, 95%CI [0.15, 1.90], p = 0.022). Additional intervention effects were also reported for improved family functioning (β = 0.64, 95%CI [0.18, 1.09], p = 0.007) and reduced caregiver mental health (β = -1.04, 95%CI [-1.90, -0.18], p = 0.018), particularly anxiety symptoms (β = -0.92, 95%CI [-1.39, -0.46], p < 0.001). No effect was detected on parenting stress or parental depression.

Sensitivity analyses adjusting for child and caregiver demographics yielded consistent findings, apart from family functioning, for which the intervention effect became statistically significant (β = 0.61, 95%CI [0.14, 1.08], p = 0.010), indicating the robustness of intervention effects (see Table 9 in S1 Text).

**Table 1. Estimated intervention effects at post-intervention (T1), based on linear, Poisson, and negative binomial multilevel regression models.**

|  | β/ IRR | SE | 95% CI | p-value |
|---|---|---|---|---|
| Early Learning and Stimulation | 1.79 | 0.79 | [0.24, 3.34] | 0.023 |
| Caregiver-Perpetrated Violence: Total | 0.87 | 0.04 | [0.80, 0.96] | 0.005 |
| Caregiver-Perpetrated Violence: Physical | 0.71 | 0.06 | [0.61, 0.84] | <0.001 |
| Caregiver-Perpetrated Violence: Emotional | 0.96 | 0.06 | [0.86, 1.08] | 0.521 |
| Attitude towards Corporal Punishment | -0.29 | 0.12 | [-0.53, -0.06] | 0.013 |
| Child Behavior: Total | -0.62 | 0.33 | [-1.27, 0.03] | 0.061 |
| Child Behavior: Internalizing behavior | -0.58 | 0.20 | [-0.97, -0.20] | 0.003 |
| Child Behavior: Externalizing behavior | -0.04 | 0.22 | [-0.47, 0.41] | 0.875 |
| Child Behavior: Emotional problem | -0.42 | 0.13 | [-0.67, -0.17] | <0.001 |
| Child Behavior: Conduct problem | -0.15 | 0.12 | [-0.40, 0.09] | 0.209 |
| Child Behavior: Hyperactivity | 0.13 | 0.17 | [-0.21, 0.47] | 0.464 |
| Child Behavior: Peer problem | -0.17 | 0.14 | [-0.43, 0.10] | 0.209 |
| Child Behavior: Prosocial behavior | -0.27 | 0.16 | [-0.58, 0.04] | 0.093 |
| Parenting Practices: Total | 2.01 | 0.69 | [0.65, 3.37] | 0.004 |
| Parenting Practices: Positive parenting | 0.99 | 0.34 | [0.33, 1.66] | 0.004 |
| Parenting Practices: Parental involvement | 1.02 | 0.45 | [0.15, 1.90] | 0.022 |
| Parental Mental Health: Total | -1.04 | 0.44 | [-1.90, -0.18] | 0.018 |
| Parental Mental Health: Depression | -0.11 | 0.27 | [-0.65, 0.42] | 0.683 |
| Parental Mental Health: Anxiety | -0.92 | 0.24 | [-1.39, -0.46] | <0.001 |
| Parenting Stress | -1.19 | 0.64 | [-2.44, 0.06] | 0.063 |
| Family Functioning | 0.64 | 0.23 | [0.18, 1.09] | 0.007 |

*Note*. IRR = incidence rate ratio; SE = standard error; CI = confidence interval.

## Disability analysis

Moderation analyses based on disability status revealed that the intervention had comparatively weaker effects on improving family functioning in families where the caregiver had a disability (β = -1.28, SE = 0.64, 95%CI [-2.54, -0.02], p = 0.047). No evidence was found that the effectiveness of the intervention differed by caregiver or child disability status across other outcomes (Tables 10a and 10b in S1 Text). However, subgroup analyses showed that, post-intervention, caregivers with disabilities appeared less likely to perpetrate total and physical violence or support corporal punishment compared to the control. Similarly, children with disabilities appeared less likely to experience caregiver-perpetrated physical violence and more likely to receive positive parenting.

## Follow-up analysis in the intervention group

All comparisons are made with reference to the baseline (T0). Compared to baseline, early learning and stimulation increased substantially at 6-month follow-up (β = 6.70, SE = 0.56, 95%CI [5.60, 7.80], p < 0.001) and 12-month follow-up (β = 2.37, SE = 0.62, 95%CI [1.14, 3.59], p < 0.001) (Fig 3; Table 11 in S1 Text). Total caregiver-perpetrated violence decreased significantly at 6-month follow-up (IRR = 0.38, SE = 0.05, 95%CI [0.34, 0.42], p < 0.001) and 12-month follow-up (IRR = 0.34, SE = 0.06, 95%CI [0.30, 0.38], p < 0.001), with similar patterns observed for both caregiver-perpetrated physical and emotional violence.

An attrition analysis was conducted to compare participants who completed the 12-month follow-up (T3) with those who did not. There were no statistically significant differences in baseline primary outcomes between

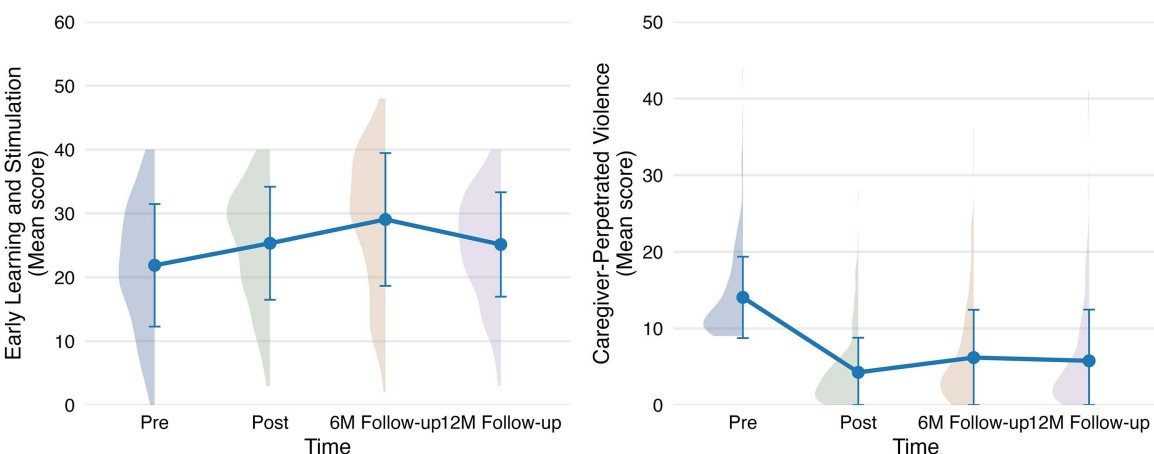

**Fig 3. Longitudinal changes in early learning and stimulation and caregiver-perpetrated violence across four timepoints.** *Note*. Each time point is accompanied by a vertical density plot. Mean values and their standard deviations are shown with points, lines, and error bars slightly offset toward the density center to enhance alignment and visual coherence.

completers and non-completers, including Early Learning and Stimulation (p = 0.063) and Caregiver-Perpetrated Violence (p = 0.984). However, statistically significant differences were observed for certain demographic characteristics, including child age, caregiver age, caregiver gender, and employment status (Table 12 in S1 Text). These findings suggest that follow-up participation may be associated with specific demographic factors, which are further considered in the Discussion.

Improvements were also observed in secondary outcomes, including child behavioral difficulties, particularly internalizing symptoms and emotional problems. Parenting practices, especially parental involvement, improved over time (Table 11 in S1 Text). Reductions in parental mental health symptoms were most evident at the 6-month follow-up. Changes in attitudes toward corporal punishment, parenting stress, and family functioning were mixed. Some gains attenuated over time, with support for corporal punishment and parenting stress showing potential increases at follow-ups.

### Predictors of engagement and causal effect among compliers

Of the 272 participants, 173 (63.6%) completed at least 30 modules and were classified as compliers as previously defined.

Analysis for baseline predictors of compliance found that higher levels of parenting stress (OR=0.94, 95%CI [0.90, 0.98], p = 0.001) and better family functioning (OR=0.90, 95%CI [0.81, 0.99], p = 0.040) were associated with lower odds of compliance. Caregivers of girls were more likely to be compliers than caregivers of boys (OR=2.09, 95%CI [1.24, 3.57], p = 0.006). Greater prosocial behavior in children (OR=1.16, 95%CI [1.02, 1.32], p = 0.023), higher levels of positive parenting practices (OR=1.04, 95%CI [1.01, 1.08], p = 0.017), and greater parental involvement (OR=1.07, 95%CI [1.01, 1.12], p = 0.016) were also positively associated with compliance (Table 13 in S1 Text).

CACE analysis revealed that, compared to the overall intention-to-treat effect, significant differences were observed in reduced endorsement of corporal punishment (β = –0.45, 95%CI [–0.78, –0.11], p = 0.011) and lower levels of anxiety (β = -0.92, 95%CI [-1.8, -0.05], p = 0.040). No differences were detected for other outcomes (Table 14 in S1 Text).

Finally, we observed considerable variation in compliance rates across clusters, ranging from 28.6% to 100% (Table 15 in S1 Text), suggesting potential contextual or implementation-related influences.

## Discussion

To our knowledge, this is the first pragmatic cluster RCT—conducted under real-world conditions within the preschool system and using broad inclusion criteria—to evaluate a universal digital-human parenting intervention for promoting child development and protection, and to assess the sustainability of impacts at 6- and 12-month follow-ups in an LMIC. It is also the first to examine disability-related equity.

At immediate post-intervention, the intervention significantly increased early learning and stimulation and reduced caregiver-perpetrated violence, particularly physical violence. Positive effects were also seen on reducing attitudes supportive of corporal punishment. These findings align with systematic reviews of non-digital selective and indicated parenting interventions that were shown to improve parent-child interaction and reduce maltreatment in children aged 0–10 [10]. Unlike those reviews, we found no effect on caregiver-perpetrated emotional violence. However, both groups decreased substantially from baseline to post-test, suggesting that broader contextual or measurement factors, such as spillover effects or heightened awareness from participation in the preschool setting, may have influenced reductions beyond the intervention itself.

The intervention significantly reduced child internalizing behaviors, especially emotional problems, consistent with evidence from in-person parenting programs [10]. However, no effect was found on overall behavioral or externalizing problems in the short term, possibly because longer time is needed for positive parenting practices to translate into changes in externalizing behaviors. Although this intervention has five chatbot modules targeting behavioral management, caregivers noted in the qualitative interviews (manuscript in preparation) that they required more intensive support to consolidate these skills. Another potential explanation is the "extinction burst", where a previously reinforced behavior is no longer rewarded, there can be a temporary increase in the intensity of that behavior before it eventually decreases, as indicated in longitudinal data [29].

Compared to waitlist control, the intervention improved parenting behaviors and caregiver mental health, particularly anxiety symptoms. These findings are consistent with research on selective and indicated digital parenting interventions [14–20,23]. No effects on depressive symptoms were observed, possibly due to a floor effect with low baseline depressive levels.

Literature shows mixed results on digital parenting interventions' effects on parenting stress. Reviews of selective and indicated interventions report reductions in stress [16,20], while those on universal interventions do not [23]. As a universal intervention, our findings align with the latter, detecting no significant between-group difference on parenting stress. Based on our qualitative inquires, possible reasons include: first, parenting stress is affected by many external factors (e.g., financial issues, co-parenting conflict) beyond the intervention's scope; second, stress measured immediately post-intervention may have temporary increases from applying new techniques, as the follow-up results did show consistent decline of parenting stress; third, despite the remote human-led interactions, the intervention still had inadequate relational buffering (e.g., peer support) to reduce stress; lastly, increased caregiver reflection on their own practices may heighten awareness of perceived inadequacies, therefore masking actual improvements.

Our exploratory subgroup analyses revealed novel findings on disability inclusion. These findings indicate that core parenting principles and skills–such as positive reinforcement, non-violent discipline, and responsive caregiving–may be effective across disability contexts. The intervention's disability-inclusive adaptations (e.g., inclusive language, framing, and visuals) may have contributed to these outcomes. However, limited improvements in other areas (e.g., family function, mental health, child behaviors) suggest the intervention was still insufficiently tailored to the unique and complex needs of families affected by disability. As such, achieving greater impacts may require a twin-track approach: delivering universal parenting support grounded in inclusive principles, alongside targeted interventions that respond to specific needs and barriers faced by families of children and caregivers with disabilities [30].

Our engagement was notably high, with 60.3% of participants completing all chatbot modules. According to the qualitative interviews with stakeholders, this success was likely driven by the trusted delivery via preschool teachers; a

child-centered token economy providing developmentally appropriate incentives to children as indirect motivation for caregivers to stay engaged; and a digital platform featuring flexible navigation, self-paced content, and human-like illustrations [31]. While digitalization and universal delivery aim for scalability, our findings reinforce that thorough needs assessment and contextual adaptation to existing systems and local culture are key to sustaining engagement.

Our CACE analysis showed no differences between participants who complied with the intervention protocol (i.e., completed 30 or more modules) and non-compliers, except in parental endorsement of corporal punishment and anxiety. Several factors may explain these findings. First, our engagement scores measured only completion of chatbot modules, which constituted a portion of the overall intervention and does not capture engagement in the group interactions. Second, high module completion rates within the chatbot may not be a reliable indication of direct engagement with parenting content. Some parents may have reflected passively going through modules to "tick the box" without internalizing or applying the content, instead of meaningful cognitive processing. Stakeholder interviews revealed that high engagement may have also been driven by frequent reminders from headteachers, who, as authority figures, may have created social pressure or fostered peer comparison among caregivers, motivating faster completion to align with group norms or maintain appearances. Third, caregivers with low chatbot use might have engaged more in other formats (e.g., group sessions), which may have driven outcomes through different mechanisms of change. Fourth, standard engagement metrics like module completion may miss qualitative aspects of engagement such as reflection, motivation, and emotional investment. Improved engagement measures are needed to account for depth, intentionality, and diversity of participation across digital and human intervention components.

Interestingly, better baseline family functioning was associated with lower odds of compliance. One possible explanation is that caregivers in more well-functioning families perceived less need for additional parenting support, leading to lower engagement intensity. This pattern may reflect differential perceived need within universal intervention settings.

Importantly, the enrolment of 25.2% male participants, achieved without targeted recruitment strategies, suggests that a digital delivery format might overcome common barriers to male engagement, such as logistical challenges and stigma. This finding highlights the need for moderator and subgroup analyses to investigate potential gender-specific effects in future research.

This study has several limitations. As outcomes relied on caregiver self-report, under-reporting due to social desirability bias cannot be excluded, particularly for sensitive behaviors such as violent discipline. Moreover, although the study included both attitudinal measures (e.g., endorsement of corporal punishment) and behavior-oriented measures (e.g., reported parenting practices and stimulation activities), these were based on self-report and therefore may reflect shifts in awareness or perceived norms rather than fully objective changes in behavior. The absence of observational or multi-informant assessments limits the ability to disentangle cognitive or attitudinal change from sustained behavioral change. However, the use of more objective observational measures was not feasible due to cost and logistical constraints, which is a common limitation in the wider literature.

Online human-led group engagement was not systematically tracked due to cultural factors, with participants potentially present but not actively interacting. The single preschool setting limits generalizability. In addition, although the waitlist control group began receiving the intervention after T1, extended follow-up of this group was not conducted due to pragmatic constraints related to the preschool academic calendar and cohort transitions. As such, sustainability effects were evaluated within the original intervention group only. Lastly, loss of control participants at follow-up, common in waitlist designs, hindered long-term effect assessment. Also, at the 12-month follow-up, attrition was associated with certain demographic characteristics. However, completers and non-completers were comparable in baseline primary outcome levels. This reduces the likelihood that sustained effects were driven solely by baseline differences. Nevertheless, long-term findings should be interpreted with consideration of the potential influence of selective retention.

This study's strengths include a rigorous cluster RCT design with inactive control, pre-registered protocol, and blinded data collection and analysis. It addressed key evidence gaps by directly measuring parenting behaviors related to early

learning and stimulation and VAC, and assessing outcome sustainability at 6- and 12-month follow-ups. The equity-focused approach, examining effects by disability status, offers novel insights into disability-inclusive universal parenting interventions.

Future research should prioritize cost-effectiveness analyses and systematically document human-led components to better quantify dosage and engagement. As cultural diversity shapes engagement, understanding cultural influences on engagement is also crucial. Future research could examine the minimum level of exposure needed to achieve sustained outcomes. More rigorous RCTs with extended follow-up periods, particularly in LMICs, are needed to strengthen evidence for universal digital parenting programs. Cross-sectoral collaboration and use of existing data platforms may enhance the monitoring of parental and child outcomes.

We underscore the importance of human-led components–whether online or offline–to enhance engagement and optimize clinical outcomes. Also, booster sessions and behavior change techniques should be incorporated to foster deeper, sustained impacts, as low-intensity or one-way interaction may be insufficient for modifying entrenched parenting behaviors and deep-seated attitudes. Moreover, digital technology should be leveraged to enable a stepped-care delivery model, combining broad public messaging to shift social norms with targeted support for high-risk groups through government–civil society partnerships. Tailoring content for intergenerational caregiving and embedding interventions in routine services with appropriate incentives can further enhance engagement and effectiveness in LMIC and oriental settings.

Early childhood development and protection is the most cost-effective equalizer to break cycles of intergenerational cycles of psychopathology and violence victimization. It is vital to seize this critical window of opportunity for all children, not only those deemed at risk, to build foundations for lifelong mental health and resilience. Our study shows that universal digital-human parenting interventions embedded in existing education systems can achieve high engagement–including among male caregivers–and yield preventative outcomes for child development and mental health at least comparable to those of selective or targeted interventions delivered digitally or in person. It supports the delivery of such interventions to expand the reach, reduce costs, increase flexibility, improve outcomes, and promote equity, thereby producing meaningful public mental health impact.

## Supporting information

**S1 File. Data.**
(XLSX)

**S1 Text. Supporting information.** Table 1: Detailed Description of Intervention Design and Delivery; Table 2: Outcome Measures; Table 3a. Minimum Detectable Effect Size (Beta) under Linear Model for Early Learning and Stimulation, Table 3b. Minimum Detectable Effect Size (IRR) under Poisson Model for Caregiver-Perpetrated Violence; Table 4. Demographic Characteristics and Baseline Measures of Participants by Group (T0); Table 5. Participant Engagement in the Chatbot; Fig 1. Distribution of Total Modules Completed Among Participants in the Intervention Group; Table 6. Means and Standard Deviations of Outcome Measures at Baseline, Immediate Post-Intervention; Table 7. Distribution Check for Caregiver-Perpetrated Violence; Table 8. Per-Protocol Sensitivity Analysis for Primary Outcomes; Table 9. Sensitivity Analyses Adjusting for Covariates: Intervention Effects on Primary and Secondary Outcomes; Table 10a. Subgroup and Moderation Analyses by Caregiver Disability Status, Table 10b. Subgroup and Moderation Analyses by Child Disability Status; Table 11. Changes in Outcome Variables from Baseline to Post-Intervention and 12-Month Follow-Up in the Intervention Group; Table 12. Baseline Characteristics by 12-Month Follow-Up Status in the Intervention Group; Table 13 Baseline Predictors of Compliance in the Intervention Group; Table 14. Estimated Complier Average Causal Effects on Primary and Secondary Outcome Variables; Table 15. Cluster-Level Compliance Rates in the Intervention Group; Methods. Pragmatic Trial Orientation; Reference.
(DOCX)

**S1 File. Protocol.**
(DOCX)

**S2 Text. Statistical analysis plan.**
(DOCX)

## Acknowledgments

We extend our sincere gratitude to Xuechen Zhang, Ruinan Zhou, Taoran Li, Xinran Liu, Qingyang Zhang, Laurie Markle, Chiara Facciolà, Wenhao Ma, Zhen Liu, Yicong Guo, Edmund Moss, Ian Stride, Lily Clements, Dongping Qiao, and Xiying Wang for their valuable contributions to the program adaptation. We are deeply grateful to Jiangxi Xinyu Chengbei Preschool, the headteachers, social workers, and participating families, whose engagement and support were essential to the successful completion of this research. We thank the LEGO Foundation and Oak Foundation for their support in the development of ParentText. **Trial Registration:** Chinese Clinical Trial Registry (part of WHO ICTRP): ChiCTR2400081911, https://www.chictr.org.cn/showproj.html?proj=221616; date of registration: March 14th, 2024.

## Author contributions

**Conceptualization:** Zuyi Fang, Cheng Zhang, Xiangming Fang.

**Data curation:** Zuyi Fang, Ruochen Ruan, Xinyu Shi, Cheng Zhang.

**Formal analysis:** Zuyi Fang, Qing Han.

**Funding acquisition:** Zuyi Fang.

**Investigation:** Zuyi Fang, Ruochen Ruan, Xinyu Shi, Cheng Zhang.

**Methodology:** Zuyi Fang, Xiangming Fang.

**Project administration:** Zuyi Fang, Ruochen Ruan, Xinyu Shi.

**Resources:** Zuyi Fang, Cheng Zhang, Dongqin Ruan, Jamie M. Lachman.

**Supervision:** Zuyi Fang, Cheng Zhang, Dongqin Ruan, Inge Vallance, Jamie M. Lachman.

**Validation:** Zuyi Fang.

**Visualization:** Qing Han.

**Writing – original draft:** Zuyi Fang, Qing Han.

**Writing – review & editing:** Zuyi Fang, Qing Han, Ruochen Ruan, Xinyu Shi, Cheng Zhang, Dongqin Ruan, Xiangming Fang, Inge Vallance, Jamie M. Lachman.

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
