## [Decision Letter · Decision Letter 0]

3 Mar 2026

Response to Reviewers'. This file does not need to include responses to any formatting updates and technical items listed in the 'Journal Requirements' section below.'. This file does not need to include responses to any formatting updates and technical items listed in the 'Journal Requirements' section below.* A marked-up copy of your manuscript that highlights changes made to the original version. You should upload this as a separate file labeled 'Revised Manuscript with Track Changes'.'.* An unmarked version of your revised paper without tracked changes. You should upload this as a separate file labeled 'Manuscript'.'. If you would like to make changes to your financial disclosure, competing interests statement, or data availability statement, please make these updates within the submission form at the time of resubmission. Guidelines for resubmitting your figure files are available below the reviewer comments at the end of this letter. We look forward to receiving your revised manuscript. Kind regards, Esli OsmanlliuHongxing LuoLeo Anthony Celi  PLOS Digital Health   **Journal Requirements:** If the reviewer comments include a recommendation to cite specific previously published works, please review and evaluate these publications to determine whether they are relevant and should be cited. There is no requirement to cite these works unless the editor has indicated otherwise.  **Additional Editor Comments (if provided):** Thank you for the opportunity to review this manuscript. The study, a pragmatic cluster RCT in a Chinese city, explores the Effectiveness of a universal digital–human parenting intervention in promoting early childhood development and protection. The study has several strengths, including the application of an adequate methodology to assess a novel intervention under real-life conditions. According to the authors, this is the first such study within the Chinese preschool system. Below are opportunities to improve the manuscript by addressing major and minor concerns:

Major:

Lines 121-129:Nicely presented knowledge gaps addressed by this study

Methods sections should precede the Results

Given the choice to offer the intervention to the control group, did the study team consider a formal cross-over design? Please justify your final design choice.

Parent input in the study seems central given the scope of the intervention. Please describe your family engagement strategy for this study.

Further describe if clusters (classrooms) were recruited before randomization, and who consented for their participation

Recommend using a tool such as the PRECIS 2 framework that scores the study across multiple design domains, and helps assess where the trial falls on the pragmatic/explanatory continuum

In the methods, describe adherence to CONSORT and relevant extension to this study design: CONSORT extensions for cluster RCTs and for pragmatic trials. Follow their recommended nomenclature and organization study components

Lines 448-452: has the PLH program been previously validated? Also provide more detail on the adaptation to Keyushiguang and any validation work in the manuscript, particularly with respect to linguistic and cultural adaptation.

What information, if any, did you collect on eligible participants who refused to participate?

How did headteacher decide on whom to invite for participation? Could this have introduced a selection bias?

Line 462 / Outcomes section: provide additional information in your manuscript on the validity and relevance of the selected scales, particularly with regards to family-centered outcomes. Focus on validation in the study setting. Also highlight who was conducting the assessments (self-report?; probability of under-reporting among abusive caregivers)

Minor:

Line 94: define what is meant by “stunting”

Lines 95-6: provide references for these claims

Line 460: “A 6-month follow-up (T2) was conducted with the intervention” ; clarify what is meant by intervention. Were all groups now receiving the intervention?

Could you also provide per protocol effect estimates as a sensitivity analysis of you primary outcome estimates?

How was cluster-level rather than participant-level noncompliance under ITT? Was this an issue in the study?

Are effects evaluated only at T1 or across all follow ups? Your model includes all time points, but the text implies T1 is the focus. Please clarify.

How were missing data handled?

Sample size: specify what intracluster correlation you assumed in your calculations

Lines 497-99: But why wasn’t the control group followed for at least 6-months post the intiation of the intervention, even if the follow-up period would have been shorter than the original intervention group?

Figure 3: clarify what the y axes represent

Line 273: “better family functioning (OR=0.90, 95%CI [0.81, 0.99], p=0.040) were associated with lower odds of compliance”. This seems counter-intuitive. How did you explain this finding?

**Reviewers' Comments:** Reviewer's Responses to Questions

**Comments to the Author**

1. Does this manuscript meet PLOS Digital Health’s publication criteria? Is the manuscript technically sound, and do the data support the conclusions? The manuscript must describe methodologically and ethically rigorous research with conclusions that are appropriately drawn based on the data presented.? Is the manuscript technically sound, and do the data support the conclusions? The manuscript must describe methodologically and ethically rigorous research with conclusions that are appropriately drawn based on the data presented.

Reviewer #1: Yes

2. Has the statistical analysis been performed appropriately and rigorously?

Reviewer #1: Yes

3. Have the authors made all data underlying the findings in their manuscript fully available (please refer to the Data Availability Statement at the start of the manuscript PDF file)?

The PLOS Data policy requires authors to make all data underlying the findings described in their manuscript fully available without restriction, with rare exception. The data should be provided as part of the manuscript or its supporting information, or deposited to a public repository. For example, in addition to summary statistics, the data points behind means, medians and variance measures should be available. If there are restrictions on publicly sharing data—e.g. participant privacy or use of data from a third party—those must be specified.requires authors to make all data underlying the findings described in their manuscript fully available without restriction, with rare exception. The data should be provided as part of the manuscript or its supporting information, or deposited to a public repository. For example, in addition to summary statistics, the data points behind means, medians and variance measures should be available. If there are restrictions on publicly sharing data—e.g. participant privacy or use of data from a third party—those must be specified.

Reviewer #1: Yes

4. Is the manuscript presented in an intelligible fashion and written in standard English?

Reviewer #1: Yes

Reviewer #1: 1. The response rate after the intervention was high at 92.8%. However, 58.1% of participants did not complete the 12-month follow-up. This loss may bias the analysis of the sustainability of the results.

2. The main outcomes, including caregiver-provided stimulation and violence against children (VAC), rely on self-reported data. This data can be affected by social desirability bias, especially when discussing sensitive topics like violence.

3. The manuscript states that 21 classes were randomized, but it does not explain the potential differences caused by variations in teachers or facilitators within those classes.

4. It is important to compare the participants who completed the 12-month follow-up with those who dropped out. This will help us understand if the "sustained effects" reflect the overall population.

5. In the Methodology section, specifically in the Data Management/Storage subsection. It is suggested to refer from the paper: “Risk Assessment for Identifying Threats, Vulnerabilities, and Countermeasures in Cloud Computing “. Since this intervention uses a chatbot and likely relies on cloud storage for caregiver data, citing Addula et al. (2025) adds important context about data safety. The referenced paper highlights weak access controls as a significant threat, accounting for 12% of incidents. This supports the need for robust "deidentified participant data" protocols outlined in the manuscript, along with effective measures such as encryption for sensitive information.

6. There is a need more information about the training and consistency of the headteachers and social workers who led the online group sessions.

7. The manuscript should better define the specific "human" elements used to encourage participants to complete the chatbot modules. This will help others replicate the study.

8. The document should consistently use terms like "caregiver-perpetrated violence" and "corporal punishment."

**Do you want your identity to be public for this peer review?** For information about this choice, including consent withdrawal, please see our Privacy Policy..

Reviewer #1: No

**Figure resubmission:**  While revising your submission, we strongly recommend that you use PLOS’s NAAS tool (https://ngplosjournals.pagemajik.ai/artanalysis) to test your figure files. NAAS can convert your figure files to the TIFF file type and meet basic requirements (such as print size, resolution), or provide you with a report on issues that do not meet our requirements and that NAAS cannot fix. 

**Reproducibility:** To enhance the reproducibility of your results, we recommend that authors of applicable studies deposit laboratory protocols in protocols.io, where a protocol can be assigned its own identifier (DOI) such that it can be cited independently in the future. Additionally, PLOS ONE offers an option to publish peer-reviewed clinical study protocols. Read more information on sharing protocols at https://plos.org/protocols?utm_medium=editorial-email&utm_source=authorletters&utm_campaign=protocols To enhance the reproducibility of your results, we recommend that authors of applicable studies deposit laboratory protocols in protocols.io, where a protocol can be assigned its own identifier (DOI) such that it can be cited independently in the future. Additionally, PLOS ONE offers an option to publish peer-reviewed clinical study protocols. Read more information on sharing protocols at https://plos.org/protocols?utm_medium=editorial-email&utm_source=authorletters&utm_campaign=protocols

---

## [Decision Letter · Decision Letter 1]

23 Mar 2026

Effectiveness of a universal digital–human parenting intervention in promoting early childhood development and protection: A pragmatic cluster randomized controlled trial

PDIG-D-25-01265R1

Dear Dr Fang,

We are pleased to inform you that your manuscript 'Effectiveness of a universal digital–human parenting intervention in promoting early childhood development and protection: A pragmatic cluster randomized controlled trial' has been provisionally accepted for publication in PLOS Digital Health.

Best regards,

Hongxing Luo, M.D., Ph.D.

Section Editor

PLOS Digital Health

**Additional Editor Comments (if provided):**

The authors are to be congratulated on this well-conducted and valuable piece of work.

I look forward to seeing more high-quality studies from this team in the future.

**Reviewer Comments (if any, and for reference):**

Reviewer's Responses to Questions

**Comments to the Author**

Reviewer #1: All comments have been addressed

publication criteria? Is the manuscript technically sound, and do the data support the conclusions? The manuscript must describe methodologically and ethically rigorous research with conclusions that are appropriately drawn based on the data presented.? Is the manuscript technically sound, and do the data support the conclusions? The manuscript must describe methodologically and ethically rigorous research with conclusions that are appropriately drawn based on the data presented.

Reviewer #1: Yes

3. Has the statistical analysis been performed appropriately and rigorously?

Reviewer #1: Yes

4. Have the authors made all data underlying the findings in their manuscript fully available (please refer to the Data Availability Statement at the start of the manuscript PDF file)?

The PLOS Data policy requires authors to make all data underlying the findings described in their manuscript fully available without restriction, with rare exception. The data should be provided as part of the manuscript or its supporting information, or deposited to a public repository. For example, in addition to summary statistics, the data points behind means, medians and variance measures should be available. If there are restrictions on publicly sharing data—e.g. participant privacy or use of data from a third party—those must be specified.requires authors to make all data underlying the findings described in their manuscript fully available without restriction, with rare exception. The data should be provided as part of the manuscript or its supporting information, or deposited to a public repository. For example, in addition to summary statistics, the data points behind means, medians and variance measures should be available. If there are restrictions on publicly sharing data—e.g. participant privacy or use of data from a third party—those must be specified.

Reviewer #1: Yes

5. Is the manuscript presented in an intelligible fashion and written in standard English?

Reviewer #1: (No Response)

Reviewer #1: No further major revisions.

**Do you want your identity to be public for this peer review?** For information about this choice, including consent withdrawal, please see our Privacy Policy..

Reviewer #1: No
